# Hospital delivery and neonatal mortality in 37 countries in sub-Saharan Africa and South Asia: An ecological study

**Anna D. Gage**[1]*, **Günther Fink**[2,3], **John E. Ataguba**[4,5], **Margaret E. Kruk**[1]

**1** Harvard T.H. Chan School of Public Health, Department of Global Health and Population, Boston, Massachusetts, United States of America, **2** Swiss Tropical and Public Health Institute, Household Economics and Health System Research Unit, Basel, Switzerland, **3** University of Basel, Basel, Switzerland, **4** Health Economics Unit, School of Public Health & Family Medicine, University of Cape Town, Cape Town, South Africa, **5** Partnership for Economic Policy, Duduville Campus, Kasarani, Nairobi, Kenya

* agage@hsph.harvard.edu

## Abstract

### Background

Widespread increases in facility delivery have not substantially reduced neonatal mortality in sub-Saharan Africa and South Asia over the past 2 decades. This may be due to poor quality care available in widely used primary care clinics. In this study, we examine the association between hospital delivery and neonatal mortality.

### Methods and findings

We used an ecological study design to assess cross-sectional associations between the share of hospital delivery and neonatal mortality across country regions. Data were from the Demographic and Health Surveys from 2009 to 2018, covering 682,239 births across all regions. We assess the association between the share of facility births in a region that occurred in hospitals (versus lower-level clinics) and early (0 to 7 days) neonatal mortality per 1,000 births, controlling for potential confounders including the share of facility births, small at birth, maternal age, maternal education, urbanicity, antenatal care visits, income, region, and survey year. We examined changes in this association in different contexts of country income, global region, and urbanicity using interaction models.

Across the 1,143 regions from 37 countries in sub-Saharan Africa and South Asia, 42%, 29%, and 28% of births took place in a hospital, clinic, and at home, respectively. A 10-percentage point higher share of facility deliveries occurring in hospitals was associated with 1.2 per 1,000 fewer deaths (p-value < 0.01; 95% CI: 0.82 to 1.60), relative to mean mortality of 22. Associations were strongest in South Asian countries, middle-income countries, and urban regions. The study's limitations include the inability to control for all confounding factors given the ecological and cross-sectional design and potential misclassification of facility levels in our data.

**Data Availability Statement:** All data files are available from the Demographic and Health Surveys Program, https://dhsprogram.com/.

**Funding:** This work was supported by funding from the Bill and Melinda Gates Foundation (INV-

005254, MK). The funder played no role in study design, data collection and analysis, decision to publish, or preparation of the manuscript.

**Competing interests:** I have read the journal's policy and the authors of this manuscript have the following competing interests: MK is a member of the Editorial Board of PLOS Medicine.

## Conclusions

Regions with more hospital deliveries than clinic deliveries have reduced neonatal mortality. Increasing delivery in hospitals while improving quality across the health system may help to reduce high neonatal mortality.

### Author summary

#### Why was this study done?

- Neonatal mortality remains high in sub-Saharan Africa and South Asia despite increasing access to facilities for delivery, potentially due to poor quality of care.

- Few studies have examined the relationship between the level of facility (hospital versus lower-level clinic) used for delivery and neonatal mortality.

#### What did the researchers do and find?

- In this ecological analysis of 1,143 subnational regions across 37 countries in sub-Saharan Africa and South Asia, we estimate the association between the share of facility births in a region that occurs in a hospital and early neonatal mortality.

- We compare this to the association between the proportion of facility deliveries in a region and the early neonatal mortality rate.

- We find that regions with higher proportions of hospital delivery have lower levels of neonatal mortality after controlling for potential confounders. We find that this association is strongest in South Asian countries (relative to sub-Saharan Africa), middle-income countries (relative to low income), and urban regions (relative to rural).

#### What do these findings mean?

- Delivery in hospitals, compared to lower-level clinics, may reduce the risk of early neonatal mortality in these 2 high mortality regions.

- A better understanding of the relationship between facility level used for delivery and mortality could guide interventions to reduce neonatal mortality in high mortality areas.

## Introduction

Despite substantial increases in facility delivery in the past 2 decades, global declines in neonatal mortality have lagged reductions in child mortality [1,2]. In 2019, of the 2.4 million neonatal deaths globally, 81% were in South Asia or in sub-Saharan Africa [3]. Despite a preponderance of evidence that good-quality delivery and newborn care can prevent most newborn deaths [4], delivering in a facility appears only weakly associated with improved

neonatal survival in these regions [5,6]. This is potentially due to the poor quality of maternity care, particularly in primary care clinics, which conduct 30% or more deliveries in low-income countries [7]. Across 5 sub-Saharan African countries, for example, quality was poorer in low-volume clinics without cesarean delivery capacity than hospitals [8]. Clinics may be unable to provide high-quality maternal and newborn care that can quickly recognize and effectively treat complications when they arise because of limited supplies and equipment, a lack of providers skilled in managing complications, and low delivery volumes, which make retaining skills challenging.

While many studies have examined the relationship between facility delivery and health outcomes [9,10], few have assessed health outcomes across facility types or levels (i.e., compared outcomes between primary care clinics and hospitals). Assessing health outcomes across facility levels is challenging in cross-sectional data because women with underlying health risks or difficulties in labor typically opt for—or are referred to—hospitals, resulting in a more challenging patient mix for hospitals [7,11]. On the other hand, hospitals tend to be located in less remote and economically more developed areas, which may lead to the opposite bias in cross-sectional analysis. To address both concerns, ecological comparisons across regions can be useful. Because populations in a region will include individuals with the full range of underlying risks, comparing health outcomes across regions with different obstetric facility use patterns should allow valid inference regarding the relative contributions of different facility levels to neonatal health outcomes.

Understanding the roles that different types of health facilities play in neonatal mortality could help to elucidate new strategies for improving maternal and newborn health. We aim to answer whether there is an association between regional hospital delivery and regional neonatal mortality. This analysis uses data from 1,143 regions in 37 countries in 2 high mortality areas, sub-Saharan Africa and South Asia, to examine the ecological relationship between the share of facility deliveries occurring in a hospital and neonatal mortality. We further examined how this relationship differs in different contexts.

## Methods

### Ethics approval

The Harvard University Human Research Protection Program categorized this secondary analysis of data as exempt from human subjects review.

### Data and settings

We used an observational ecological study design encompassing all countries in sub-Saharan Africa and South Asia with available data. This study did not have a prospective analysis plan. We compiled data from the most recent Demographic and Health Survey (DHS) [12] for each country in these 2 areas available after 2000 [13]. DHS surveys are nationally representative population surveys that include questions about healthcare utilization and child mortality for children born to women of reproductive age in selected households. Surveys from 37 countries collected between 2009 and 2018 met the inclusion criteria; the countries and survey dates are listed in Table 1. For each survey, we identified the lowest administrative level for which the survey was intended to be representative, most often the region or province level. We defined each of the variables of interest at the child level for children born 5 years before the survey, then used the household sampling weights to collapse the dataset to a regional level.

**Table 1. Study countries and place of delivery.**

| Country | DHS year | N regions | Area | Income group | Place of birth | | | |
|---|---|---|---|---|---|---|---|---|
| | | | | | Hospital | Clinic | Home | Unknown facility |
| Afghanistan | 2015 | 34 | South Asia | Low | 40% | 8% | 51% | 1% |
| Angola | 2015 | 18 | sub-Saharan Africa | Middle | 35% | 10% | 53% | 1% |
| Bangladesh | 2014 | 7 | South Asia | Middle | 37% | 1% | 62% | 0% |
| Benin | 2017 | 12 | sub-Saharan Africa | Low | 30% | 53% | 14% | 2% |
| Burkina Faso | 2010 | 13 | sub-Saharan Africa | Low | 6% | 61% | 33% | 0% |
| Burundi | 2017 | 18 | sub-Saharan Africa | Low | 21% | 63% | 12% | 4% |
| Cameroon | 2011 | 12 | sub-Saharan Africa | Middle | 34% | 27% | 38% | 1% |
| Chad | 2015 | 21 | sub-Saharan Africa | Low | 10% | 12% | 78% | 0% |
| Congo, Dem. Rep. | 2014 | 11 | sub-Saharan Africa | Low | 26% | 48% | 19% | 7% |
| Congo, Rep. | 2012 | 12 | sub-Saharan Africa | Middle | 68% | 21% | 8% | 0% |
| Côte d'Ivoire | 2012 | 11 | sub-Saharan Africa | Middle | 23% | 34% | 41% | 1% |
| Ethiopia | 2016 | 11 | sub-Saharan Africa | Low | 7% | 18% | 73% | 2% |
| Gabon | 2012 | 10 | sub-Saharan Africa | Middle | 77% | 10% | 7% | 4% |
| Ghana | 2014 | 10 | sub-Saharan Africa | Middle | 53% | 20% | 27% | 0% |
| Guinea | 2012 | 8 | sub-Saharan Africa | Low | 14% | 26% | 59% | 0% |
| India | 2016 | 634 | South Asia | Middle | 50% | 29% | 21% | 0% |
| Kenya | 2014 | 47 | sub-Saharan Africa | Middle | 39% | 16% | 38% | 1% |
| Lesotho | 2014 | 10 | sub-Saharan Africa | Middle | 59% | 14% | 23% | 4% |
| Liberia | 2013 | 5 | sub-Saharan Africa | Low | 35% | 20% | 44% | 1% |
| Madagascar | 2009 | 22 | sub-Saharan Africa | Low | 8% | 26% | 64% | 2% |
| Malawi | 2016 | 28 | sub-Saharan Africa | Low | 33% | 56% | 7% | 4% |
| Mali | 2018 | 9 | sub-Saharan Africa | Low | 5% | 61% | 33% | 1% |
| Mozambique | 2011 | 11 | sub-Saharan Africa | Low | 19% | 33% | 43% | 3% |
| Namibia | 2013 | 13 | sub-Saharan Africa | Middle | 83% | 5% | 12% | 1% |
| Nepal | 2016 | 7 | South Asia | Low | 40% | 13% | 41% | 5% |
| Niger | 2012 | 8 | sub-Saharan Africa | Low | 6% | 23% | 70% | 1% |
| Nigeria | 2013 | 37 | sub-Saharan Africa | Middle | 27% | 8% | 63% | 0% |
| Pakistan | 2017 | 6 | South Asia | Middle | 64% | 1% | 34% | 1% |
| Rwanda | 2015 | 5 | sub-Saharan Africa | Low | 27% | 64% | 8% | 1% |
| Senegal | 2017 | 14 | sub-Saharan Africa | Low | 15% | 63% | 21% | 2% |
| Sierra Leone | 2013 | 4 | sub-Saharan Africa | Low | 13% | 40% | 44% | 1% |
| Swaziland | 2007 | 4 | sub-Saharan Africa | Middle | 66% | 8% | 25% | 1% |
| Tanzania | 2016 | 30 | sub-Saharan Africa | Low | 32% | 30% | 36% | 2% |
| Togo | 2014 | 6 | sub-Saharan Africa | Low | 29% | 44% | 27% | 1% |
| Uganda | 2016 | 15 | sub-Saharan Africa | Low | 36% | 37% | 25% | 2% |
| Zambia | 2014 | 10 | sub-Saharan Africa | Middle | 20% | 44% | 31% | 5% |
| Zimbabwe | 2015 | 10 | sub-Saharan Africa | Low | 39% | 31% | 20% | 10% |

## Variables

The primary outcome is the region's early neonatal mortality rate (death within 7 days of birth) per 1,000 births. We also considered the neonatal mortality rate (within 28 days of birth) per 1,000 births. Both outcomes are strongly influenced by the quality of care around the time of delivery [4]. As a sensitivity test, we also examined postneonatal mortality rate (between 29 days and 1 year), which should not be directly influenced by the delivery facility but the broader health system and the social determinants of health. Stillbirths were included

as early neonatal deaths if they were reported by the mother as deaths occurring on the day of birth; however, they were not explicitly considered as an outcome.

The explanatory variable of interest is the share of births occurring in a hospital in each region among those occurring in any facility, whether a hospital, lower-level clinic, or unclassified facility. Hospitals were first defined using the country's own definitions, while clinics include all non-hospitals, i.e., dispensaries, health centers, or doctors' offices. A small percentage of births occur in an unclassified facility type. Given that there is no universal definition of a hospital across countries, we then further validated the accuracy of the hospital/clinic categorization by assessing the proportion of cesarean deliveries in each, as this service is not typically available in lower-level clinics and often signals the presence of higher-level capabilities including an operating room, an obstetrician, and blood. S1 Table presents the percent of births delivered by cesarean section by facility level and country. A category of clinics (i.e., "Upazila health complex" in Bangladesh) was recategorized as "hospital" in a country if over 10% of deliveries in that category were cesarean deliveries. No facilities were recategorized from hospital to clinic. As a reference point for the effect sizes, we also defined the percent of facility births as the sum of the hospital, clinic, and unknown facility births, as opposed to home births. Following reviewer feedback, we also conducted a robustness check in which the primary explanatory variable was defined as the share of births occurring in a hospital in each region among all births (including home births).

We identified a set of covariates associated with both birth location and neonatal mortality to control for potential confounding. Included covariates needed to be available from all DHS surveys in our sample, have known associations with neonatal mortality [14,15], are determined and knowable prior to delivery, and are relevant at the population rather than the individual level. We first controlled for facility births in the hospital share models. We also included in all adjusted models the percent of babies in the region that were small at birth (<2,500 grams or mother's report of smaller than average); percent of multiple births; average maternal age at birth; percent of first births; percent of births with a preceding interval less than 2 years (for non-first births), the median number of antenatal care visits (only asked among most recent births); the percent of mothers with only primary education and with secondary or higher education; the percent of households that were in an urban area; the average estimated annual household income as estimated by the International Center for Equity in Health; the global region (sub-Saharan Africa or South Asia); the country's income level (low or middle); and an indicator for the survey year [16]. The country's income category is drawn from the World Bank Group classifications; given the small number of countries in the upper-middle-income category (just Namibia and Gabon), we combined upper-middle- and low-middle-income categories into a single middle-income group. Maternal anemia, which is also associated with perinatal and neonatal mortality, was assessed in only 28 of the 37 countries in our sample [17]. We, therefore, conducted a robustness check that also adjusted for maternal anemia prevalence in this subset of countries following reviewer feedback. An initial set of covariates included percent male and excluded maternal age and percent of first births; this was revised following reviewer feedback.

If a variable was missing at the child level, the region's average was estimated without that child's data, effectively imputing the missingness to be the regional average. Missingness for all variables of interest at the child level is shown by country in S2 Table. Most variables had low levels of missingness, except for small at birth in Kenya. We conduct a robustness check excluding Kenya from the analysis to account for the reduced variation in this covariate.

## Analysis

We summarized the variables of interest for the study regions; the countries in each region are listed in Table 1. We used unadjusted and adjusted 2-level random intercept models, with

regions nested within countries, to estimate separately the associations of regional share of hospital delivery and regional facility delivery on regional neonatal mortality outcomes. Adjusted models included the full set of covariates listed above. We did not correct for multiple testing in the main analysis; Bonferroni-adjusted *p*-values to account for the 3 different outcomes are presented in S3 Table [18]. Based on these initial results, we looked at 3 different contextual variables to understand how these associations may vary across contexts: low- versus middle-income countries, sub-Saharan African versus South Asian countries, and urban versus rural regions (urban defined as a population that is >50% urban). We used adjusted 2-level random intercept interaction models where hospital delivery or facility delivery share was interacted with the relevant contextual variable. All models were adjusted using the same covariates as listed above; the full model specification is included in S4 Table. We used these models to plot margins by predicting early neonatal mortality across the range of hospital or delivery share holding all other covariates at their mean values, and varying the level of the contextual variable. Robustness checks that interact the contextual variable with all the model covariates are included in the Supporting information [19]. A fourth contextual variable, time of the survey (during or before 2015 versus during or after 2016), was further considered in the Supporting information. Finally, after reviewing the contextual variables, we further explored the associations in sub-Saharan Africa with a random slope and random intercept model, allowing the relationship between hospital delivery and early neonatal mortality to vary by country.

This study is reported as per the Strengthening the Reporting of Observational Studies in Epidemiology (STROBE) guideline (S1 Checklist). Analyses were conducted in Stata version 15.1. The Harvard University Human Research Protection Program approved this secondary data analysis as exempt from human subjects review.

## Results

A total of 682,239 births across the 1,143 study regions and 37 countries were included in the analysis; summary statistics are presented in Table 2. The average early neonatal mortality rate was 22 deaths per 1,000 births, 42% of births took place in a hospital, 29% took place in a clinic, and 28% took place at home. Hospital births made up 49% of all facility deliveries in sub-Saharan Africa and 64% in South Asia. S5 Table shows the within-country variation in place of delivery, which, for most countries, was substantial; in Ethiopia, for example, the share of hospital deliveries among facility births was 16% in the minimum region and 73% in the maximum region, while the share of facility deliveries ranged from 15% to 97%.

A higher share of facility deliveries occurring in hospitals was associated with lower early neonatal deaths and neonatal deaths (Table 3) after controlling for covariates, while share of delivery in any facility was associated with higher mortality (Table 4). A 10-percentage point increase in hospital delivery among facility births was associated with 1.2 fewer early neonatal deaths per 1,000 births (*p*-value < 0.01; 95% CI: 0.83 to 1.58), or a 5% decrease relative to the mean early neonatal mortality rate. In contrast, a 10-percentage point increase in the share of any facility deliveries (as compared to home births) was associated with 0.89 increase in early neonatal mortality (*p*-value = 0.02; 95% CI: 0.17,1.61). As expected, delivery location had a lower association with postneonatal deaths than with early neonatal deaths. S6 Table presents the unadjusted models. S7 Table shows the exposure defined as the share of hospital births among all births: a 10-percentage point increase in overall hospital deliveries is associate with 1.58 fewer early neonatal deaths (*p*-value < 0.01; 95% CI 1.23,1.93). These associations are robust to the inclusion of anemia prevalence as a covariate in a subset of countries and the exclusion of Kenya from the analysis (S8 and S9 Tables).

**Table 2. Characteristics of sample regions.**

| | Sub-Saharan Africa | South Asia | Overall |
|---|---|---|---|
| **Data** | | | |
| N countries | 32 | 5 | 37 |
| N regions | 455 | 688 | 1,143 |
| Outcomes | | | |
| Early neonatal death per 1,000 | 23 | 22 | 22 |
| Neonatal death per 1,000 | 26 | 26 | 26 |
| Postneonatal death per 1,000 | 22 | 11 | 15 |
| Place of delivery | | | |
| Hospital mean % | 31 | 49 | 42 |
| Lower facility mean % | 30 | 27 | 29 |
| Home mean % | 36 | 23 | 28 |
| Unknown facility mean % | 2 | 0 | 1 |
| Covariates | | | |
| Small at birth mean % | 19 | 21 | 20 |
| Median number of antenatal care visits | 3.9 | 4.1 | 4.0 |
| Multiple births mean % | 4 | 2 | 2 |
| Maternal age mean | 26.9 | 25.2 | 25.9 |
| Urban mean % | 29 | 25 | 27 |
| First birth mean % | 23 | 38 | 32 |
| Less than 2 year birth interval mean % | 17 | 18 | 17 |
| Mother primary education only mean % | 39 | 14 | 24 |
| Mother secondary education or higher mean % | 27 | 57 | 45 |
| Average annual income[a] (USD) *Mean* | 8,907 | 13,101 | 11,431 |

Early neonatal deaths are deaths between 0–7 days; neonatal deaths are deaths between 0–28 days; postneonatal deaths are deaths 29–365 days. Unclassified facility is where we could not determine the level (hospital vs. nonhospital). Small at birth is birth weight <2,500 grams or mother's report at birth. Mother's primary education is completed only primary education; mother's secondary education or higher is completed secondary or higher education. Number of antenatal care visits only defined for woman's most recent birth.

[a]Average annual income based on estimates from the International Center for Equity in Health; they are in 2011 international dollars adjusted at purchasing power parity [16].

Fig 1 shows the marginal relationship between place of delivery and early neonatal mortality by contextual variables. Hospital delivery was protective in regions in middle-income countries, in South Asia, and in both urban and rural regions, but not within low-income countries or in sub-Saharan Africa. However, facility delivery share was not protective in any of the groups of regions examined. The full interaction models are shown in S4 Table. S1 Fig and S10 Table show the interaction by year of survey; hospital delivery was more protective in later surveys than in 2015 or before. Results from the random intercept and slope models from the sub-Saharan African region are in S11 Table; the share of hospital delivery was more protective for early neonatal mortality in Namibia and Kenya and less protective in Lesotho and Côte d'Ivoire. The results are robust to including the contextual variable and covariate interactions (S2 Fig and S12 Table).

## Discussion

This analysis of place of facility delivery and neonatal mortality in the highest mortality regions of the world found that subnational regions that had a higher share of facility births in

**Table 3. Associations between share of hospital delivery and deaths per 1,000 births in 1,143 regions.**

| | Early neonatal death (per 1,000 births) | | | Neonatal death (per 1,000 births) | | | Postneonatal death (per 1,000 births) | | |
|---|---|---|---|---|---|---|---|---|---|
| | Coef. | p-value | 95% CI | Coef. | p-value | 95% CI | Coef. | p-value | 95% CI |
| Hospital % among facility deliveries | −12 | 0.00 | [−15.8,−8.3] | −14.9 | 0.00 | [−20.0,−9.8] | −4.4 | 0.04 | [−8.6,−0.2] |
| All facility % | 6.3 | 0.04 | [0.2,12.4] | 7 | 0.06 | [−0.3,14.3] | −6 | 0.16 | [−14.4,2.3] |
| Small at birth % | 6.6 | 0.07 | [−0.6,13.7] | 10.5 | 0.01 | [2.8,18.3] | 2.6 | 0.63 | [−8.1,13.3] |
| Antenatal care visit median | −0.4 | 0.04 | [−0.8,−0.0] | −0.6 | 0.01 | [−1.0,−0.1] | −0.2 | 0.22 | [−0.6,0.1] |
| Multiple birth % | 16.4 | 0.00 | [11.8,20.9] | 18.7 | 0.00 | [15.2,22.2] | 4.4 | 0.05 | [0.1,8.8] |
| Average maternal age | −0.1 | 0.71 | [−0.7,0.5] | −0.1 | 0.82 | [−0.7,0.6] | 0.1 | 0.64 | [−0.3,0.6] |
| Urban % | 0.2 | 0.93 | [−3.3,3.6] | 0.1 | 0.94 | [−3.2,3.5] | 1.9 | 0.09 | [−0.3,4.2] |
| First birth % | −8.4 | 0.19 | [−21.1,4.3] | −9.3 | 0.16 | [−22.2,3.7] | −8 | 0.40 | [−26.5,10.5] |
| Less than 2-year birth interval % | 21 | 0.03 | [2.5,39.4] | 26.3 | 0.02 | [4.0,48.7] | 20.1 | 0.01 | [5.8,34.5] |
| Mother's primary education % | 2.1 | 0.54 | [−4.6,8.7] | 0.5 | 0.88 | [−6.7,7.8] | 3.1 | 0.41 | [−4.2,10.5] |
| Mother's secondary education or higher % | −13.6 | 0.00 | [−21.1,−6.0] | −16.3 | 0.00 | [−24.6,−7.9] | −1.4 | 0.34 | [−4.2,1.4] |
| Average annual income | −0.1 | 0.93 | [−2.0,1.8] | 0.7 | 0.38 | [−0.9,2.4] | −1 | 0.44 | [−3.5,1.5] |
| South Asia (vs. sub-Saharan Africa) | 5.6 | 0.11 | [−1.3,12.6] | 7.6 | 0.06 | [−0.4,15.6] | −0.3 | 0.92 | [−5.6,5.1] |
| Middle-income country (vs. low-income) | 6.3 | 0.00 | [2.0,10.6] | 6.6 | 0.01 | [1.9,11.2] | −3.2 | 0.10 | [−7.1,0.7] |
| N | 1143 | | | 1143 | | | 1143 | | |

Models also include survey year fixed effects for years 2009–2018. Small at birth is birth weight <2,500 grams or mother's report at birth. Number of antenatal care visits only defined for woman's most recent birth. Mother's primary education is completed only primary education; mother's secondary education or higher is completed secondary or higher education. Log average annual income based on estimates from the International Center for Equity in Health; they are in log 2011 international dollars adjusted at purchasing power parity.

**Table 4. Associations between share of deliveries in any facility and deaths per 1,000 births in 1,143 regions.**

| | Early neonatal death (per 1,000 births) | | | Neonatal death (per 1,000 births) | | | Postneonatal death (per 1,000 births) | | |
|---|---|---|---|---|---|---|---|---|---|
| | Coef. | p-value | 95% CI | Coef. | p-value | 95% CI | Coef. | p-value | 95% CI |
| All facility % | 8.9 | 0.02 | [1.7,16.1] | 10.3 | 0.02 | [1.5,19.1] | −4.9 | 0.25 | [−13.1,3.4] |
| Small at birth % | 6.3 | 0.09 | [−1.1,13.7] | 10.3 | 0.01 | [2.3,18.2] | 2.5 | 0.67 | [−8.7,13.6] |
| Antenatal care visit median | −0.6 | 0.00 | [−1.0,−0.2] | −0.8 | 0.00 | [−1.3,−0.3] | −0.3 | 0.09 | [−0.6,0.0] |
| Multiple birth % | 16.9 | 0.00 | [12.6,21.2] | 19.3 | 0.00 | [16.2,22.4] | 4.6 | 0.04 | [0.2,8.9] |
| Average maternal age | −0.3 | 0.41 | [−0.9,0.4] | −0.3 | 0.47 | [−0.9,0.4] | 0.1 | 0.80 | [−0.4,0.6] |
| Urban % | −1.6 | 0.41 | [−5.3,2.1] | −2 | 0.31 | [−5.8,1.8] | 1.4 | 0.24 | [−0.9,3.6] |
| First birth % | −13.5 | 0.04 | [−26.4,−0.6] | −15.6 | 0.02 | [−28.6,−2.7] | −9.9 | 0.28 | [−28.1,8.2] |
| Less than 2-year birth interval % | 15.4 | 0.08 | [−1.6,32.5] | 19.8 | 0.05 | [−0.0,39.7] | 18.5 | 0.02 | [3.2,33.8] |
| Mother's primary education % | 1 | 0.77 | [−5.9,8.0] | −0.4 | 0.92 | [−7.8,7.0] | 3.1 | 0.41 | [−4.4,10.6] |
| Mother's secondary education or higher % | −15.3 | 0.00 | [−23.2,−7.3] | −18.3 | 0.00 | [−27.0,−9.6] | −1.9 | 0.18 | [−4.6,0.9] |
| Average annual income | −1.9 | 0.23 | [−5.1,1.2] | −1.6 | 0.34 | [−4.8,1.6] | −1.8 | 0.07 | [−3.7,0.1] |
| South Asia (vs. sub-Saharan Africa) | 4.1 | 0.32 | [−3.9,12.0] | 5.7 | 0.22 | [−3.4,14.9] | −0.7 | 0.81 | [−6.4,5.0] |
| Middle-income country (vs. low-income) | 5 | 0.06 | [−0.2,10.1] | 4.9 | 0.09 | [−0.8,10.5] | −3.8 | 0.07 | [−7.9,0.4] |
| N | 1,143 | | | 1,143 | | | 1,143 | | |

Models also include survey year fixed effects for years 2009–2018. Small at birth is birth weight <2,500 grams or mother's report at birth. Number of antenatal care visits only defined for woman's most recent birth. Mother's primary education is completed only primary education; mother's secondary education or higher is completed secondary or higher education. Log average annual income based on estimates from the International Center for Equity in Health; they are in log 2011 international dollars adjusted at purchasing power parity.

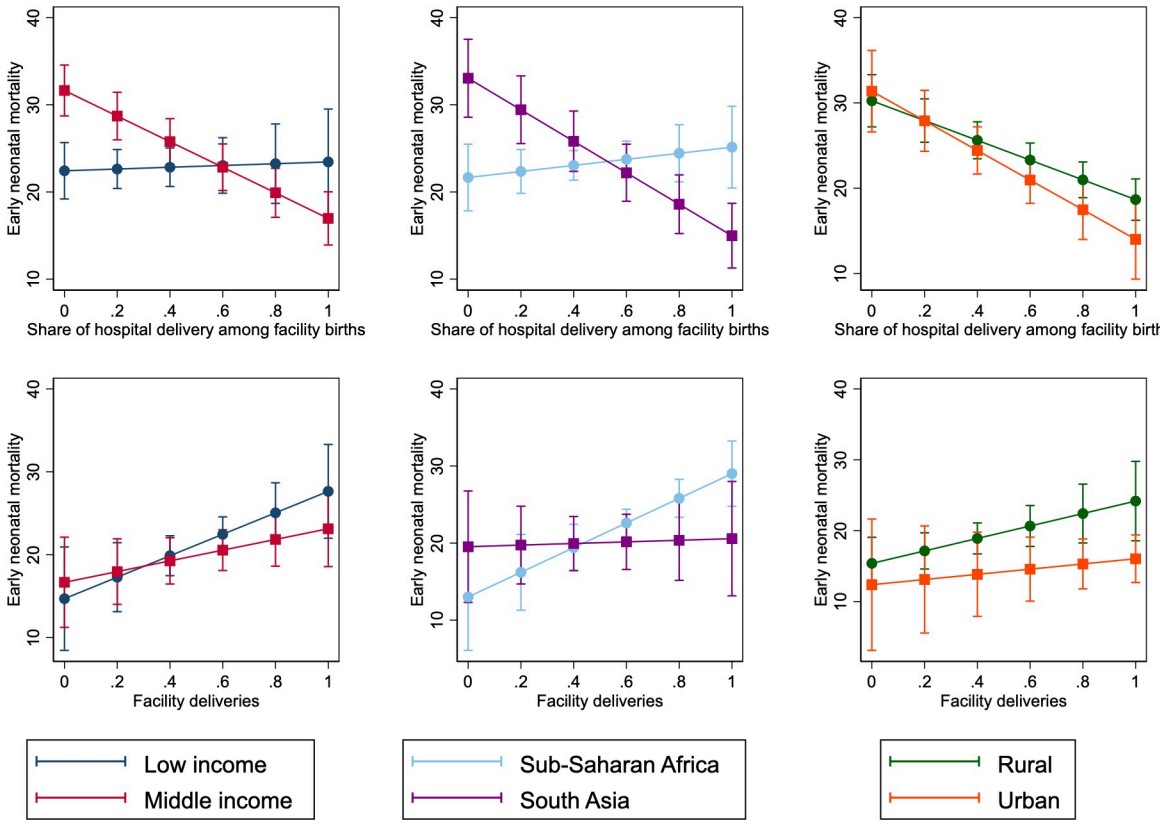

**Fig 1. Early neonatal mortality and place of delivery by country income, country area, and urban region.** Each graph shows a different multivariable random intercept model that is fully adjusted. Points show the marginal estimate of early neonatal mortality for the given level of hospital deliveries or facility deliveries; bars show 95% confidence intervals. Country income groups defined by the World Bank Group, with low-middle- and upper-middle-income countries combined. Rural regions defined as those with less than 50% of the population classified as urban, urban regions defined as the converse. Mortality defined as death within 7 days per thousand births.

hospitals had lower early neonatal mortality, controlling for income and sociodemographic factors. Regions where all facility births occurred in hospitals had an estimated 12.1 fewer early neonatal deaths per 1,000 births than regions with all facility delivery in clinics. By contrast, a greater proportion of births in all facilities was associated with greater newborn mortality in our analysis. These associations were seen most strongly in regions in South Asia and middle-income countries. The associations did not differ between urban and rural regions; hospital delivery was protective in both.

Hospitals may be protective against mortality than other facilities because they have more advanced equipment, infrastructure, and health workers to manage emergencies if they arise; higher-volume facilities may also allow greater practice in managing facilities [8,20]. Regions with high hospital delivery may also have stronger health systems, including more health financing or better transportation, facilitating both hospital delivery and lower mortality. These hypotheses may account for the different associations seen across contexts. First, the quality of hospitals may be higher in middle-income countries than in low-income countries, making delivery in them more protective. Cesarean section capacity is more common in middle-income country hospitals than in hospitals in low-income countries; care for small or sick newborns may also be more advanced in middle-income country hospitals [21–23]. Second, a greater proportion of health spending is pooled in middle-income countries in comparison to low-income countries, which may remove financial barriers to hospital delivery for a greater percentage of

women [24]. As a majority of the low-income regions were in countries in sub-Saharan Africa, these reasons may also explain the differences between South Asia and sub-Saharan Africa. Across all the categories examined, however, the share of hospital delivery among facility deliveries was more protective than the share of facility delivery. Furthermore, we found that while the unadjusted relationship between any facility delivery and mortality was negative, once we controlled for confounders, particularly regional income, secondary education and urbanicity, any facility delivery share was actually positively associated with neonatal mortality. Facility delivery was found to be only weakly associated with neonatal mortality reductions in the literature [5,6].

The estimated size of the association between hospital delivery and early neonatal delivery is comparable to those seen in other studies examining the level or quality of facility conducting deliveries. An analysis from Malawi found that higher-quality facilities (the majority of which were hospitals) were associated with a 23 fewer neonatal deaths per 1,000 births than other facilities in the country [25]; similarly, in Ghana, reduction in stillbirth was only found among women who live near higher-quality CEmONC facilities but not lower-level BEmONC facilities [26]. Relocation of care to high-quality facilities following a transportation intervention in Ghana was also found to reduce facility-based maternal mortality [27]. Some interventions have also found similar levels of reductions in mortality: A study from Rwanda also found a 12.9 deaths per 1,000 reduction in neonatal mortality following an intensive quality improvement intervention in 2 rural districts [28], while a neonatal resuscitation intervention in Nepal reduced intrapartum-related mortality by 3 deaths per 1,000 [29]. However, many other health system–oriented neonatal health interventions have failed to reduce mortality [30–32]. New strategies such as supporting hospital delivery should therefore be explored further.

There are several limitations to this analysis. First, although we attempted to control for confounding and conduct sensitivity tests, this is an observational and ecological analysis that may not have been able to control for all confounding; for example, areas with more hospitals may also have more staff or supplies per facility. Second, we relied on women's self-report of the level of facility where they delivered. Women may not know the level of the facility or use the term "hospital" to colloquially mean a health facility; furthermore, there are no global definitions of what constitutes a hospital, meaning that the exposure differs from country to country. We addressed this by conducting random and fixed effects models with regions nested within countries, but there may still be within-country variations in these definitions. It is likely that our hospital category includes smaller, nonsurgical facilities, which would bias our hospital association downward. Third, while we hypothesize that improved quality of care in hospitals may be a key factor for the observed associations, limited cross-country data on quality of care prevent direct measurement. Finally, this analysis did not consider stillbirths as an outcome because they were inconsistently recorded in versions of the DHS that relied on a full birth history rather than a full pregnancy history [33]; neonatal deaths by cause were also not considered as that information is unavailable.

Efforts to improve primary care facilities have proliferated in the past 2 decades; despite this, newborn mortality remains stubbornly high in many regions [2,34]. A recent large randomized controlled trial found that a package of interventions to improve delivery care at primary care facilities in India failed to reduce maternal and newborn mortality in the absence of onsite operative delivery and other advanced services [30]. Shifting deliveries out of primary care clinics and into facilities that have advanced services to manage complications and sufficient delivery volume to maintain clinical expertise has been proposed [35]. To be effective, such a policy would require additional investment in hospitals to ensure excellent lifesaving and respectful midwifery and obstetric care, improving the quality of antenatal and postnatal care quality in clinics, decreasing geographic and financial barriers to care, and igniting population demand for high-quality maternity care [35]. Prospective evaluation using

implementation science methods can provide needed evidence on how to best achieve success-ful health system redesign, in which contexts health system redesign may be most beneficial, and how to mitigate unintended consequences.

Health system innovations are required to meet the ambitious targets in the Sustainable Development Goals for neonatal mortality (12 deaths/1,000 live births). As countries work to meet these goals, structural redesign of health systems that places a premium on high-quality obstetric care is a potential opportunity to reduce mortality that warrants further exploration.

## Supporting information

**S1 STROBE Checklist. Strengthening the Reporting of Observational Studies in Epidemi-ology (STROBE) checklist.**
(DOC)

**S1 Table. C-section rates by facility level and country.**
(DOCX)

**S2 Table. Percent of births missing key variables by country.**
(DOCX)

**S3 Table. Bonferroni-adjusted models.**
(DOCX)

**S4 Table. Interaction models specification and full model results.**
(DOCX)

**S5 Table. Within-country variation in place of delivery.**
(DOCX)

**S6 Table. Unadjusted main model results.**
(DOCX)

**S7 Table. Association between percent of all deliveries in hospital and neonatal mortality.**
(DOCX)

**S8 Table. Robustness check including anemia as covariate in subset of study countries.**
(DOCX)

**S9 Table. Robustness check excluding Kenya from analysis.**
(DOCX)

**S10 Table. Interaction model comparing associations in early versus late surveys.**
(DOCX)

**S11 Table. Random intercept and slopes in sub-Saharan Africa.**
(DOCX)

**S12 Table. Interaction models including contextual variable and covariate interactions.**
(DOCX)

**S1 Fig. Interaction model comparing associations in early versus late surveys.** Each graph shows a different multivariable random intercept model that is fully adjusted. Points show the marginal estimate of early neonatal mortality for the given level of hospital deliveries or facility deliveries; bars show 95% confidence intervals. Mortality defined as death within 7 days per thousand births.
(TIF)

**S2 Fig. Interaction models including contextual variable and covariate interactions.** Each graph shows a different multivariable random intercept model that is fully adjusted with all covariates as well as interactions between each of the covariates and contextual variable. Points show the marginal estimate of early neonatal mortality for the given level of hospital deliveries or facility deliveries; bars show 95% confidence intervals. Country income groups defined by the World Bank Group, with low-middle- and upper-middle-income countries combined. Rural regions defined as those with less than 50% of the population classified as urban, urban regions defined as the converse.
(TIF)

## Author Contributions

**Conceptualization:** Anna D. Gage, Günther Fink, John E. Ataguba, Margaret E. Kruk.

**Formal analysis:** Anna D. Gage.

**Methodology:** Günther Fink, John E. Ataguba, Margaret E. Kruk.

**Visualization:** Anna D. Gage, Margaret E. Kruk.

**Writing – original draft:** Anna D. Gage.

**Writing – review & editing:** Günther Fink, John E. Ataguba, Margaret E. Kruk.

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
