## [Editor Report · Decision Letter 0]

25 May 2021

Dear Dr Gage, 

Thank you for submitting your manuscript entitled "The association between hospital delivery and neonatal mortality: an ecological analysis in 37 countries in sub-Saharan Africa and South Asia" for consideration by PLOS Medicine.

Your manuscript has now been evaluated by the PLOS Medicine editorial staff and I am writing to let you know that we would like to send your submission out for external peer review.

Please re-submit your manuscript within two working days, i.e. by May 27 2021 11:59PM.

Kind regards,

Beryne Odeny

Associate Editor

PLOS Medicine

---

## [Decision Letter · Decision Letter 1]

2 Aug 2021

Dear Dr. Gage,

Thank you very much for submitting your manuscript "The association between hospital delivery and neonatal mortality: an ecological analysis in 37 countries in sub-Saharan Africa and South Asia" (PMEDICINE-D-21-02269R1) for consideration at PLOS Medicine. 

[LINK]

In light of these reviews, I am afraid that we will not be able to accept the manuscript for publication in the journal in its current form, but we would like to consider a revised version that addresses the reviewers' and editors' comments. Obviously we cannot make any decision about publication until we have seen the revised manuscript and your response, and we plan to seek re-review by one or more of the reviewers. 

We expect to receive your revised manuscript by Aug 23 2021 11:59PM. Please email us (plosmedicine@plos.org) if you have any questions or concerns.

We look forward to receiving your revised manuscript. 

Sincerely,

Beryne Odeny, 

PLOS Medicine 

plosmedicine.org

1) Please revise your title according to PLOS Medicine's style. Your title must be nondeclarative and not a question. It should begin with main concept if possible. Please place the study design ("A ecological study,") in the subtitle (i.e., after a colon). For example, “Hospital delivery and neonatal mortality in 37 countries in sub-Saharan Africa and South Asia: An ecological study”

2) Abstract:

a) Please structure your abstract using the PLOS Medicine headings (Background, Methods and Findings, Conclusions).

b) Please combine the Methods and Findings sections into one section, “Methods and findings”. Please ensure that all numbers presented in the abstract are present and identical to numbers presented in the main manuscript text.

c) Please include the actual amounts or percentages of relevant outcomes, not just hazard ratios.

d) Please include the important dependent variables that are adjusted for in the analyses.

e) Please include p values in addition to 95% CIs.

f) In the last sentence of the Abstract Methods and Findings section, please describe the main limitation(s) of the study's methodology.

3) Author summary - At this stage, we ask that you reformat your non-technical Author Summary. The Author Summary should immediately follow the Abstract in your revised manuscript. This text is subject to editorial change and should be distinct from the scientific abstract. The summary should be accessible to a wide audience that includes both scientists and non-scientists. Please see our author guidelines for more information: https://journals.plos.org/plosmedicine/s/revising-your-manuscript#loc-author-summary.

4) Please conclude the Introduction with a clear description of the study question or hypothesis.

5) Did your study have a prospective protocol or analysis plan? Please state this (either way) early in the Methods section. 

i) If a prospective analysis plan (from your funding proposal, IRB or other ethics committee submission, study protocol, or other planning document written before analyzing the data) was used in designing the study, please include the relevant prospectively written document with your revised manuscript as a Supporting Information file to be published alongside your study, and cite it in the Methods section. A legend for this file should be included at the end of your manuscript. 

ii) If no such document exists, please make sure that the Methods section transparently describes when analyses were planned, and when/why any data-driven changes to analyses took place. 

iii) In either case, changes in the analysis-- including those made in response to peer review comments-- should be identified as such in the Methods section of the paper, with rationale.

6) Please ensure that the study is reported according to the STROBE guideline, and include the completed STROBE checklist as Supporting Information. Please add the following statement, or similar, to the Methods: "This study is reported as per the Strengthening the Reporting of Observational Studies in Epidemiology (STROBE) guideline (S1 Checklist)." The STROBE guideline can be found here: http://www.equator-network.org/reporting-guidelines/strobe/

7) In your statistical analyses, please use hierarchical/ multilevel models given that nationwide data is likely clustered at various county/ regional levels. The potential clustering of data (e.g., among patients from the same locality or hospital) would result in spurious effect estimates and standard errors. Did you use any approaches to reduce bias due to cluster‐level confounders?

8) In statistical methods, please refer to any post-hoc corrections to correct for multiple comparisons during your statistical analyses. If these were not performed, please justify the reasons. 

9) Please describe how you selected your adjustment variables. 

10) In the methods section, please provide both 95% CIs and p values in the text.

11) Please specify the statistical test used to derive the p values.

12) Please include p-values in all tables.

13) Please define the abbreviations in tables and figures e.g. ANC

14) Please indicate in the figure caption the meaning of the bars and whiskers in Figure 1

15) Discussion section, line 13, should be “particular” and not “particularly”

16) Please use the "Vancouver" style for reference formatting and see our website for other reference guidelines https://journals.plos.org/plosmedicine/s/submission-guidelines#loc-references.

a) Please use the PLOS Medicine style reference call outs throughout the text, noting the absence of spaces within the square brackets, e.g., "... child mortality [1,2]."

b) Please ensure that journal name abbreviations consistently match those found in the National Center for Biotechnology Information (NCBI) databases. https://journals.plos.org/plosmedicine/s/submission-guidelines#loc-references. 

c) Please ensure that all weblinks are current and accessible to date.

d) Please remove “[dataset]” from ref #12

17) Please include line numbers in the next draft

Comments from the reviewers:

Reviewer #1: "The association between hospital delivery and neonatal mortality: an ecological analysis in 37 countries in sub-Saharan Africa and South Asia" presents an positive association between hospital delivery and reduced neonatal mortality, in particular that a 10% higher share of hospital deliveries was associated with a 6% decrease in early neonatal mortality, or 1.21 per 1000 fewer deaths, from an ecological study design on DHS data from some 680,000 births over 1,143 regions in 37 countries in sub-Saharan Africa and South Asia.

These findings appear not in total agreement with prior studies on much the same DHS source data cited as [5,6], with [6] in particular investigating the same broad regions of sub-Saharan Africa and South Asia also through ecological analyses. These prior work suggest weak/non-significant associations between institutional deliveries/health facilities and neonatal mortality (e.g. from Figure 2B in [6]). The main methodological distinction for this paper appears to be in recognizing different tiers/qualities of health institutions/facilities, in particular between "hospitals" (with capability to perform advanced procedures such as Caesarean deliveries) and "primary-care clinics" (with fewer such capabilities). Increased significance could then be found for neonatal mortality once the comparison is based on (higher-tier) hospital delivery, as suggested from Table S2, rather than on all facilities (which appears correlated with early neonatal births; Table 4). These associations however appear mixed when analyzed by contextual variables (e.g. Figure 1), and between countries (e.g. Table S4).

Overall, the main conclusions appear plausible, and of importance in motivating better quality of maternal/natal care, especially in less-developed countries. The methodological design also appears appropriate. However, the various analyses might stand to be described in greater detail to facilitate potential replication, and there remain some concerns over the validity of the hospital/clinic definitions used:

1. In Page 5, it is stated that "The explanatory variable of interest is the share of births occurring in a hospital in each region among those occurring in any facility, whether a hospital, lower-level clinic, or unclassified facility" (i.e. hospital/all facilities); is there a reason why it was not "share of births occuring in a hospital in each region among all births whether in a facility or not" (i.e. hospital/all births)? This might be considered as a secondary analysis.

2. In Page 5, it is stated that "We validated the accuracy of the hospital/clinic categorization by assessing the proportion of Caesarean deliveries in each, as this service is not typically available in lower-level clinics". It might be explained in much greater detail as to how this validation influenced the categorization. For example, was there some threshold of Caesarean deliveries above which the facility would be categorized for purposes of the analyses as a hospital, and otherwise a clinic, and if so, what threshold? And if so, was a single universal threshold applied, since it is later stated that "Caesarean section capacity may in particularly be more common in middle income country hospitals, than in hospitals in low income countries"? Or were facilities not matching their self-definition ignored? In either case, by what extent did any re-categorization differ from the initial categorization, or if no re-categorization was done, by what extent did the original self-categorization differ from the expected categorization from Caesarean deliveries?

3. Related to the above, since the ability to perform Caesarean deliveries (and related advanced services) appears a major factor suggested for the protectiveness of hospital delivery (e.g. "Across five sub-Saharan African countries, for example, quality was poorer in low-volume clinics without Caesarean delivery capacity than hospitals", Page 3), it might be considered to perform brief statistical presentation/analyses relating specifically to Caesarean delivery, if at all possible.

For example, while it is stated that "we relied on women's self-report of the level of facility where they delivered. Women may not know the level of the facility or use the term 'hospital' to colloquially mean a health facility" (Page 9), might it be determined as to whether the facilities mentioned did or did not have Caesarean delivery capability, and perform brief analyses according to that firm criteria? Or alternatively perform sensitivity analyses on only facilities for which the status of such capability is known for certain?

4. From Page 5, the set of covariates appears relatively limited, and many reasonable population-level demographic/health confounders apparently not included (e.g. average age of mother at birth, average prevalence of various common diseases/conditions known to affect neonatal mortality, etc.); these might be discussed more comprehensively, if possible.

5. It would be highly encouraged to provide more details (e.g. full specification, equations, covariates adjusted for, coefficients found, etc.) about the various analyses and models (two-level random intercept, interaction [also by contextual variables], etc.), as mentioned on Page 6 and Page 7, possibly in the supplementary material. Some details seem already provided (e.g. Table S3 for the interaction model), and might be expanded upon.

6. The treatment of stillbirths in these analyses might be explicitly stated, since it appears a significant consideration in prior related studies.

Reviewer #2: Thank you very much for inviting me to review this interesting piece of secondary analysis of demographic health survey in low and middle income countries. 

Researchers aim to explore the difference in neonatal mortality (early or late) by the place of birth and provide explanation on what might have been the possible reasons. Researchers have adjusted the possible confounders to assess the association with neonatal mortality and have come to the conclusion that the mortality is high in clinic or primary health care center. From the 32 countries in Sub-saharan Africa and South Asia, using the DHS dataset, 1143 regions were analyzed. The mortality in these 1143 regions were analyzed and also the proportion of hospital birth, primary health care unit and home was analysed. The estimation of the neonatal mortality rate in region was then associated using a linear regression model with the proportion of birth in hospital, primary health care unit and home. 

To adjust the confounders, maternal education, average annual income, urban/rural, birth interval, multiple birth, antenatal care visit and sex of baby was done. 

This is an interesting way of analyzing the DHS which collects data at individual level using a PPS method and extracting data at sub-national level or regional level to assess mortality with place of birth. I have two major observations and two major feedback to the work

1. The regional mortality estimate does not take into consideration the obstetric complication that is the major factor for early neonatal mortality. As for the late neonatal mortality, there are other postnatal exposures such as infection practices and care of small and sick newborns. I know this is difficult to assess however, DHS does provide information on complication during pregnancy and delivery based on self reporting. Can this been done

2. The regionalized mortality estimate of each region and facility birth takes into a fixed model consideration that all women go to the hospital directly from home, and there is no intra-facility referral in place, which increases the risk of mortality. I know this is not provided in DHS, but a proxy measure is available in DHS questionnaires.

3. The high mortality in primary health facilities is postulated, yet not shown in the data might have been due to poor staffing pattern and competency in health facility. Using the three delay care model, the delay in arrival to health facility is one of the major factor for mortality in first referral units. Can the researchers analyse the birth preparedness indicator.

4. There has been recent global debates on whether all births should take place in hospitals and primary health care facility should be at outreach clinic, which ANC and PNC by , Roder-DeWan and colleagues in BMJ Global Health. As shown by Ashish KC and colleagues in Journal of global health (Perfect Storm, 2021), that quality of intrapartum care is associated with number of facility birth. I think this has not been considered.

Reviewer #3: This is a critically important topic and although DHS data have limitations in answering the question, this is a well conducted analysis and the adjusted models have controlled for a reasonable set of confounders.

Which countries introduced of voucher schemes or free maternity care, and when? Could this be incorporated in the model? The rationale for this is that introduction of such schemes has commonly increased the number of facility-based deliveries without the corresponding provision of staffing and resources to cope with the increased numbers.

A '10 percentage point increase in hospital delivery among facility births' is not quite the same thing as suggested by the article title 'The association between hospital delivery and neonatal mortality'. All facility births are compared to home births, for completeness I think the same needs to be presented for hospital births.

In the same light, the sentence in the Discussion 'Across all the categories examined however, share of hospital delivery was more protective than share of facility delivery' really should read 'Across all the categories examined however, share of hospital delivery among facility deliveries was more protective than the overall share of facility delivery'.

Please state how much data were missing. Imputing the missingness to be the regional average artificially reduces the width of the distribution and thus potentially increases power of analyses. This is not much of an issue if missingness is rare, but if more than 5 to 10%, could have impacted the findings.

The analysis is a strong approach, however using DHS data does not allow us to know whether the facility births were those of the small size babies, multiple births etc. Nor were there data on the quality of facilities or quality improvement schemes. Some of these are mentioned in the discussion, but lack of such data is a limitation that should be mentioned. A potential bias is that stillbirths may not be reported as early neonatal mortality at facilities and both stillbirths and early neonatal deaths may also occur at home and not be reported in national statistics or at DHS rounds (https://pophealthmetrics.biomedcentral.com/articles/10.1186/s12963-020-00225-0).

The argument seems strongest for reducing births at small facilities where resources, staffing and training may be limited - this point is well made in the discussion.

Reviewer #4: Thank you for submitting this interesting article on the association between hospital delivery and neonatal mortality.

As you state in your introduction, there have been substantial increases in the proportion of facility-based deliveries, without the corresponding decreases in neonatal mortality. It's important to try to understand why this is the case. This article provides some insight into a potential reason for this.

The following are some general comments/suggestions:

Abstract:

You should state in the background section that you are examining the association between place of delivery and neonatal mortality. Specifically, you seem to be interested in whether delivering a baby in the hospital, clinic or at home is associated with a reduction neonatal mortality. You should also state in the abstract the basis for your analysis and your hypothesis - quality of care in primary care clinics may be sub-optimal.

What you state in your conclusions is not reflecting your findings. Indeed, your findings suggests that deliveries at hospital are associated with a reduction in early neonatal mortality rates. The data you have obtained don't have any information on the quality of care at these hospitals. Future research should try to understand why tertiary level facilities had lower early neonatal mortality rates.

Introduction:

You need to describe what constitutes high quality maternal and newborn care.

You need to ensure that your objective is accurately stated before your methods - whether neonatal mortality is associated with place of delivery including primary, lower level clinics and home deliveries.

Methods

* Substantial changes in the proportion of women delivering at health facilities took place during data collection. You should consider adjusting for this in your analysis, or perhaps stratifying on different time periods.

* The same applies to improving the quality of care between 2009 and 2018, so I feel stratification on time may be warranted.

* Some variables you have included as confounders are not correct. All confounders should be variables that take place before delivery, and not after delivery. If you select variables that include after delivery, they will be on the causal pathway. So the following variables are not appropriate: small at birth, multiple births, male babies. If the DHS data has information on difficulties in the antenatal period, this may be a relevant confounder. Also, previous neonatal deaths, stillbirths, miscarriages could be considered.

* During the time period that data was collected, women could have more than one delivery. It is important to account for this in the analysis. Perhaps some type of GEE model?

* You should include the statistical software you used for your analysis as well as the commands you used for your models. It seems as though you used marginal models. What command did you use for these models? 

* This is an important paper you should probably reference: https://www.ncbi.nlm.nih.gov/pmc/articles/PMC5463806/

Minor comments: 

Some copy editing is required.

Your discussion includes a recommendation of using implementation science methods to achieve health system redesign. Could you elaborate further how implementation science could be used to help improve the quality of care at health facilities? 

Reviewer #5: The authors have presented a comprehensive analysis of the association between hospital delivery and neonatal mortality in 37 countries in sub-Saharan Africa and South Asia. The manuscript is detailed, well-written, and has the potential to make a valuable contribution to the literature. I have one minor comment about the limitations of the study. A good number of babies born at health facility dies at the facility without leaving (the facility). Unfortunately, the DHS questionnaires do not single these deaths out. Besides, to be effective in improving neonatal survival in those countries, adding the causes of deaths module to the survey instruments can provided much needed credence to the conclusions of this study.

[LINK]

---

## [Decision Letter · Decision Letter 2]

22 Sep 2021

Dear Dr. Gage,

Thank you very much for re-submitting your manuscript "Hospital delivery and neonatal mortality in 37 countries in sub-Saharan Africa and South Asia: An ecological study" (PMEDICINE-D-21-02269R2) for review by PLOS Medicine.

I have discussed the paper with my colleagues and the academic editor and it was also seen again by three reviewers. I am pleased to say that provided the remaining editorial and production issues are dealt with we are planning to accept the paper for publication in the journal.

[LINK]

We look forward to receiving the revised manuscript by Sep 29 2021 11:59PM.   

Sincerely,

Beryne Odeny, 

Associate Editor 

PLOS Medicine

plosmedicine.org

Requests from Editors:

1) Author summary: Please trim the content under “What did the researchers do and find?” to 2-3 sentences per bullet point.

2) Please remove the ‘Funding’, “Data availability statement” and “Conflict of interest” from the end of the main text. In the event of publication, this information will be published as metadata based on your responses to the submission form.

3) Please define the abbreviation, ANC, in the table footnotes 

4) Please provide the meaning of the bars and whiskers in Figure S1.

5) Please label the Y axis in figure S1 more clearly, i.e., early neonatal mortality (deaths per 1000) or similar\\

6) References: Please ensure that journal name abbreviations consistently match those found in the National Center for Biotechnology Information (NCBI) databases

Comments from Reviewers:

Reviewer #1: We thank the authors for considering our previous suggestions, and note that the various additional sensitivity analyses generally support the initial claims. A couple of points remain for consideration:

1. The interaction model specification in S4 Table is much appreciated. There is however a concern that the covariates should be further controlled for potential alternative interactions (see: https://towardsdatascience.com/interaction-analyses-appropriately-adjusting-for-control-variables-d34dfbdd781a); this might be considered as another robustness check.

2. The formatting of Tables 3 and 4 might however be reconsidered in the latest manuscript, since the final column for post-neonatal death appears to be partially cut off.

Reviewer #2: The researchers have addressed most of the comments made and highlighted the limitation on examining the quality of care from DHS. However, can the immediate newborn care practice such as skin to skin contact and immediate breast feeding be used as a proxy to quality of care. 

Reviewer #3: My comments have been adequately addressed.

[LINK]

---

## [Decision Letter · Decision Letter 3]

8 Oct 2021

Dear Dr Gage, 

On behalf of my colleagues and the Academic Editor, Dr. Jenny E Myers, I am pleased to inform you that we have agreed to publish your manuscript "Hospital delivery and neonatal mortality in 37 countries in sub-Saharan Africa and South Asia: An ecological study" (PMEDICINE-D-21-02269R3) in PLOS Medicine.

PRESS

Sincerely, 

Beryne Odeny 

PLOS Medicine